# SPFormer: Enhancing Vision Transformer with Superpixel Representation

## Abstract

In this study, we present a novel approach to enhance Vision Transformers by leveraging superpixel representation. Unlike the traditional Vision Transformer, which uniformly partitions images into non-overlapping patches of fixed size, our superpixel approach divides an image into distinct, irregular regions, each designed to cluster pixels based on shared semantics for better capturing intricate image details. Note that this superpixel clustering is also applicable at the intermediate feature level. The resulting model, denoted as SPFormer, can be trained end-to-end, and empirically demonstrates superior performance across a range of benchmarks. Additionally, SPFormer provides better interpretability through the visualization of its learned superpixels, and exhibits strong robustness against challenging testing conditions like rotation and occlusion.

## 1 Introduction

Over the past decade, the vision community has witnessed a remarkable evolution in visual recognition systems, from the resurgence of Convolutional Neural Networks (CNNs) in 2012 (Krizhevsky et al., 2012) to the cutting-edge innovation of Vision Transformers (ViTs) in 2020 (Dosovitskiy et al., 2021). This progression has instigated a significant shift in the underlying methodology for feature representation learning, transitioning from pixel-based (for CNNs) to patch-based (for ViTs).

Conventionally, pixel-based representations organize an image as a regular grid, allowing CNNs (He et al., 2016; Tan & Le, 2019; Mehta & Rastegari, 2022; Sandler et al., 2018) to extract local detailed features through sliding window operations. Despite the inductive bias inherent in CNNs, like translation equivariance, aiding their success in effectively learning visual representations, these networks face a challenge in capturing global-range information, typically necessitating the stacking of multiple convolutional operations and/or additional operations (Li et al., 2020; Chen et al., 2018) to enlarge their receptive fields.

On the other hand, ViTs (Dosovitskiy et al., 2021) regard an image as a sequence of patches. These patch-based representations, usually of a much lower resolution compared to their pixel-based counterparts, enable global-range self-attention operations in a computationally efficient manner. While the attention mechanism successfully captures global interactions, it does so at the expense of losing local details, like object boundaries. Moreover, the low resolution of patch-based representations poses challenges to adaptation for high-resolution dense prediction tasks such as segmentation and detection, which require both local detail preservation and global context information.

This leads us to ponder an interesting question: *can we derive benefits from both preserved local details and effective long-range relationship capture*? In response, we explore superpixel-based solutions, which have been employed extensively in computer vision prior to the deep learning era (Zhu & Yuille, 1996; Shi & Malik, 2000; Martin et al., 2001; Malik et al., 2001; Borenstein & Ullman, 2002; Tu & Zhu, 2002; Ren & Malik, 2003). These solutions provide locally coherent structures and reduce computational overhead compared to pixel-wise processing. Specifically, superpixels partition an image into irregular regions, with each region grouping pixels with similar semantics. This approach allows for a small number of superpixels, making it amenable to modeling global interactions through self-attention.

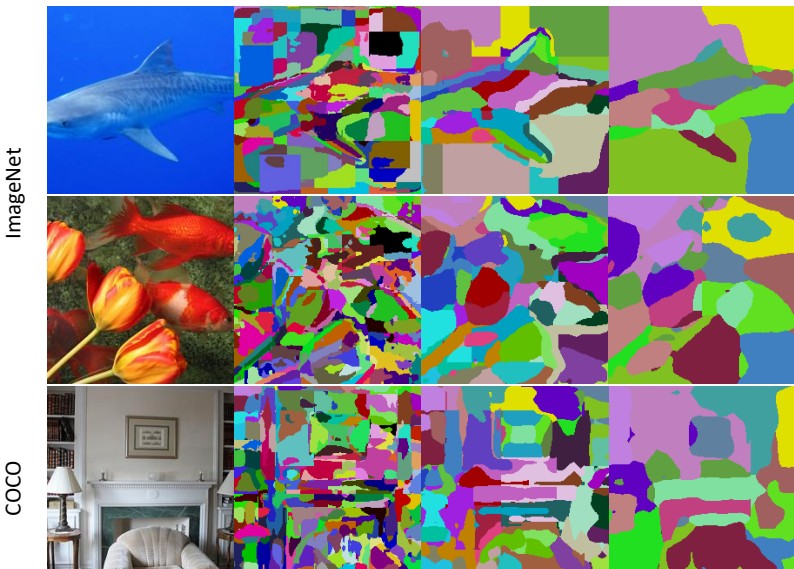

Figure 1: Visualization of learned superpixels with our SPFormer trained on ImageNet with category labels only. For each row, we show input image, visualization of 196, 49, and 16 superpixels. The learned superpixel aligns well with the object boundaries even with 16 superpixels. The last row shows results from a COCO image (not trained), demonstrating SPFormer's zero-shot ability.

Motivated by this, we propose to enhance ViTs by transitioning from patch representation to super-pixel representation via the proposed Superpixel Cross Attention (SCA). The resulting architecture, dubbed Superpixel Transformer (SPFormer), preserves image local details while facilitating global-range self-attention, and can be trained end-to-end. Compared to the vanilla ViT architecture, our SPFormer exhibits a substantial performance boost on the challenging ImageNet benchmark, *e.g.*, with gains of 1.4% for ViT-T and 1.1% for ViT-S. Furthermore, the learned superpixel representations exhibit better robustness to rotation and occlusion, transferability to open-vocabulary data to some extent, and also provide a higher level of interpretability. With these encouraging results, we hope our work will inspire the community to explore representation beyond pixels and patches.

## 2   RELATED WORK

**Pixel Representation**   Convolutional Neural Networks (CNNs) (LeCun et al., 1998; Krizhevsky et al., 2012; Simonyan & Zisserman, 2015; Szegedy et al., 2015; Ioffe & Szegedy, 2015; He et al., 2016; Tan & Le, 2019; Liu et al., 2022) process an image as a grid of pixels in a sliding window manner. CNN has been the dominant network choice since the advent of AlexNet (Krizhevsky et al., 2012), benefiting from several design choices, such as translation equivariance and the hierarchical structure to extract multi-scale features. However, it requires stacking several convolution operations to capture long-range information (Simonyan & Zisserman, 2015; He et al., 2016), and it could not easily capture global-range information, as the self-attention operation (Vaswani et al., 2017).

**Patch Representation**   The self-attention mechanism (Bahdanau et al., 2015) of Transformer architectures (Vaswani et al., 2017) effectively captures long-range information. However, its computation cost is quadratic to the number of input tokens. Vision Transformers (ViTs) (Dosovitskiy et al., 2021) alleviate the issue by tokenizing (or patchifying) the input image with a sequence of patches (*e.g.*, patch size $16 \times 16$). The patch representation (Dosovitskiy et al., 2021) unleashes the power of Transformer architectures (Vaswani et al., 2017) in computer vision, significantly impacting multiple visual recognition tasks (Russakovsky et al., 2015; Carion et al., 2020; Zhu et al., 2021; Wang et al., 2021a; Touvron et al., 2021a; Radford et al., 2021; Bao et al., 2022; He et al., 2022; Yu et al., 2022c). Due to the lack of the built-in inductive biases as in CNNs, learning with ViTs requires special training enhancements, *e.g.*, large-scale datasets (Sun et al., 2017), better training recipes (Touvron et al., 2021a; Steiner et al., 2021), or architectural designs (Liu et al., 2021; Wang et al., 2021b). To mitigate the issue, a few works exploit convolutions (LeCun et al., 1998; Sandler

et al., 2018) to tokenize the images, resulting in hybrid CNN-Transformer architectures (Wu et al., 2021; Yuan et al., 2021; Dai et al., 2021; Xiao et al., 2021; Mehta & Rastegari, 2022; Guo et al., 2022; Tu et al., 2022; Yang et al., 2023). Unlike those works that simply gather knowledge from existing CNNs and ViTs, we explore a different superpixel representation in vision transformers.

**Superpixel Representation**   Before the deep learning era, superpixel is one of the most popular representations in computer vision (Zhu & Yuille, 1996; Shi & Malik, 2000; Martin et al., 2001; Malik et al., 2001; Borenstein & Ullman, 2002; Tu & Zhu, 2002; Ren & Malik, 2003). Ren and Malik (Ren & Malik, 2003) preprocess images with superpixels that are locally coherent, preserving the structure necessary for the following recognition tasks. It also significantly reduces the computation overhead, compared to the pixel-wise processing. The superpixel clustering methods include graph-based approaches (Shi & Malik, 2000; Felzenszwalb & Huttenlocher, 2004), mean-shift (Comaniciu & Meer, 2002; Vedaldi & Soatto, 2008), or k-means clustering (Lloyd, 1982; Achanta et al., 2012). Thanks to its effective representation, recently some works attempt to incorporate clustering methods into deep learning frameworks (Jampani et al., 2018; Yang et al., 2020; Locatello et al., 2020; Xu et al., 2022; Yu et al., 2022a; Zhang et al., 2022; Yu et al., 2022b; Ma et al., 2023; Huang et al., 2023). For example, SSN (Jampani et al., 2018) integrates the differentiable SLIC (Achanta et al., 2012) to CNNs, allowing end-to-end training. Yu et al. (2022a;b) regard object queries (Carion et al., 2020; Wang et al., 2021a) as cluster centers in Transformer decoders (Vaswani et al., 2017). SViT (Huang et al., 2023) clusters the tokens to form the super tokens, where the clustering process has no gradient passed through[1]. Consequently, their network is not aware of the clustering process and could not recover from the clustering error. CoCs (Ma et al., 2023) groups pixels into clusters, while aggregating features within each cluster by regarding the image as a set of points with coordinates concatenated. In contrast, our proposed method groups pixels into superpixels, and models their global relationship via self-attention. Furthermore, during clustering, CoCs uses a Swin-style window partition (Liu et al., 2021) that introduces visual artifacts, especially around the window boundaries.

## 3   METHOD

We first formalize the adopted superpixel representation, and compare it with other conventional alternatives in Sec 3.1. Following this, we instantiate the superpixel representation with the proposed Superpixel Cross Attention (SCA) in Section 3.2. Empowered by the superpixel representation, we then propose our model SPFormer in Section 3.3.

### 3.1   SUPERPIXEL REPRESENTATION

**Pixel Representation**   A high-resolution pixel representation of an image $I \in \mathcal{R}^{c \times h \times w}$ is considered as a regular grid of pixels, where $c, h, w$ denote the number of channels, height, width, respectively. It is typically processed by CNNs with the sliding window operation.

**Patch Representation**   A low-resolution patch representation of an image $P \in \mathcal{R}^{c \times ph \times pw}$ significantly reduces the input length from thousands of pixels to a few hundreds of patches, where $ph, pw$ is the number of patches in height and width, respectively. The small input length of $P$ is thus more tractable for self-attention and is commonly processed by Transformer networks. However, the use of patch representation often incurs a loss of details due to its coarse granularity. Note that applying self-attention directly to the high-resolution image $I$ would be computationally intractable, given that the complexity of self-attention scales quadratically with the number of pixels.

**Superpixel Representation**   The adopted superpixel representation consists of the superpixel features $S_f \in \mathcal{R}^{c \times sh \times sw}$ and the association $S_a \in \mathcal{R}^{n \times h \times w}$ between pixels and superpixels, where $n$ is the number of neighboring superpixels used to represent a pixel, and throughout this paper, we consistently set $n$ to 9. $sh$ and $sw$ denote the number of superpixels in height and width, respectively.

The superpixel representation can be easily cast into the pixel representation $I \in \mathcal{R}^{c \times h \times w}$ as follows:

$$I(i) = \sum_{p \in \mathcal{N}(i)} S_a(i, p) * S_f(p) , \tag{1}$$

---

[1] https://github.com/hhb072/STViT/blob/main/models/stvit.py#L190

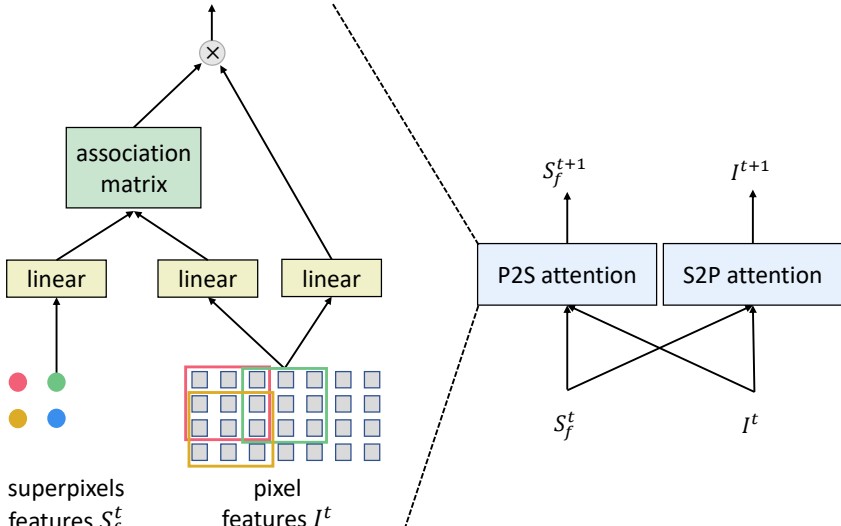

Figure 2: To achieve iterative refinement of both superpixel features and pixel features, we adopt a sliding window-based cross-attention mechanism (each superpixel cross-attends to a small local region of pixels, highlighted in the color rectangle). In the left figure, we provide a detailed illustration of the Pixel-to-Superpixel (P2S) cross-attention, while the Superpixel-to-Pixel (S2P) cross-attention follows a similar approach by reversing the roles of superpixel and pixel.

where $\mathcal{N}(i)$ are the neighboring superpixels $p$ for pixel $i$. Basically, to recover the pixel feature $I(i)$ for each $i$, we first find its neighboring superpixels $\mathcal{N}(i)$ and project the superpixel features $S_f$ back to pixel feature via the association $S_a$.

The superpixel representation offers multiple benefits. By maintaining a lower resolution (*i.e.*, fewer superpixels), it allows for efficient modeling of global interactions via self-attention. Moreover, pixels with similar semantics are learned to be grouped into the same superpixel, encoding detailed information such as object boundaries. Furthermore, superpixel provides a more robust representation than patch to rotation and occlusion, as the superpixel is adjustable w.r.t. image distortion. Consequently, the superpixel representation is **efficient** (by having lower resolution than pixel representation and even patch representation), **explainable** (formed by grouping pixel features with similar semantics), and **robust** to rotation and occlusions, as demonstrated in the experiments.

## 3.2 SUPERPIXEL CROSS ATTENTION (SCA)

Given an initial pixel representation $I^0$ and superpixel features $S_f^0$, we perform the following iterative updates: At each iteration $t$, we update the superpixel features $S_f^t$ and the association $S_a^t$ using cross-attention in a sliding window manner, as illustrated in Figure 2. Particularly, the proposed Superpixel Cross Attention (SCA) module contains two cross-attentions: Pixel-to-Superpixel (P2S) and Superpixel-to-Pixel (S2P) cross-attention. We adopt a sliding window-based cross-attention mechanism to preserve the locality of superpixels and maintain high efficiency. In the P2S cross-attention, each superpixel feature will cross-attend to a local region of pixel features, while in the S2P cross-attention, the roles of superpixel and pixel features are swapped. The proposed SCA module allows us to progressively enhance the superpixel representations and establish precise pixel-to-superpixel assignments (*i.e.*, associations). We provide more details below.

To encode position information within SCA, we adopt convolution position embedding (CPE) (Huang et al., 2023) to capture spatial relationships within the image. Before applying the P2S and S2P cross-attentions, both the superpixel and pixel features are separately enriched by CPE, which is instantiated by a $3 \times 3$ depthwise convolution (with skip connection) following Huang et al. (2023). The enriched superpixel and pixel features fortify the association between pixels and superpixels based on their relative positions. This spatial awareness encourages pixels to associate with superpixels in their close proximity.

In the P2S cross-attention, we update the superpixel features $S_f^t$ by aggregating pixel features within its sliding window. Specifically, we compute the updated superpixel features $S_f^t(p)$ as follows:

$$S_f^t(p) = S_f^{t-1}(p) + \sum_{i \in \mathcal{N}(p)} \text{softmax}_i \left( q_{S_f^{t-1}(p)} \cdot k_{I^{t-1}(i)} \right) v_{I^{t-1}(i)}, \tag{2}$$

where $\mathcal{N}(p)$ represents the set of pixels surrounding superpixel $p$. The query $(q)$, key $(k)$, and value $(v)$ are obtained through linear transformations of the previous superpixel features $S_f^{t-1}(p)$ and pixel features $I^{t-1}(i)$ (as denoted in the underscript).

In the S2P cross-attention, we update the pixel features $I^t$ using the neighboring superpixels. Before that, we first assign each pixel $i$ to its corresponding superpixel $p$ by computing the association scores $S_a^t(i, p)$ as follows:

$$S_a^t(i, p) = \text{softmax}_{p \in \mathcal{N}(i)} \left( q_{I^{t-1}(i)} \cdot k_{S_f^{t-1}(p)} \right), \tag{3}$$

where $\mathcal{N}(i)$ is the neighboring superpixels of pixel $i$. The query $(q)$ and key $(k)$ are obtained through linear transformations of the previous pixel features $I^{t-1}(i)$ and superpixel features $S_f^{t-1}(p)$ (as denoted in the underscript).

Subsequently, the pixel representation $I^t$ is updated via the updated association scores as follows:

$$I^t(i) = I^{t-1}(i) + \sum_{p \in \mathcal{N}(i)} S_a^t(i, p) \cdot v_{S_f^{t-1}(p)}, \tag{4}$$

where the value $(v)$ is obtained through the linear transformation of the previous superpixel features $S_f^{t-1}(p)$ (as denoted in the underscript). These update equations ensure that the pixel representations are refined based on the associated superpixel features, thereby enhancing the overall quality of the feature representation. One proposed SCA module can refine both the pixel and superpixel features with $t$ iterations. With the proposed SCA module in mind, we now introduce the proposed architecture SPFormer in the following subsection.

### 3.3 SPFORMER ARCHITECTURE

In order to simplify the analysis of the proposed superpixel representation and minimize modifications from the vanilla ViT (Dosovitskiy et al., 2021), we have made minimal adjustments. Following the approach of ViT, we utilize a non-overlapping patchify layer but with a smaller window size $4 \times 4$ to extract initial pixel features. Our method enables a smaller window size $4 \times 4$, instead of $16 \times 16$, thanks to the effective superpixel representation that can further effectively reduce the input length.

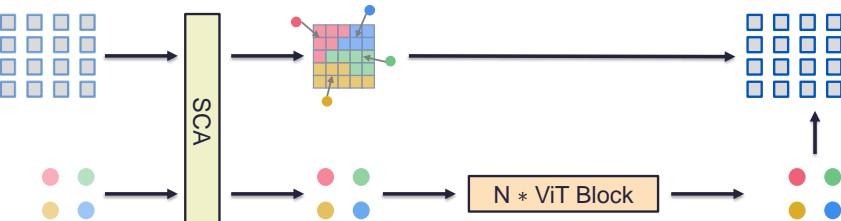

Figure 3: Architecture of a SPFormer stage. Minimal operations are directly performed on the dense pixel representation, delegating most of the computation to our efficient superpixel representation.

The superpixel features $S_f^0$ are initialized using a $1 \times 1$ convolution and $4 \times 4$ average pooling based on the pixel features $I^0$. We employ Superpixel Cross Attention (SCA) to update the superpixel features $S_f^0$ and the association $S_a^0$ (with $t$ iterations), as depicted in Sec 3.2. SCA leverages the local spatial context within a superpixel to enhance its representation. After the SCA module, we apply multi-head self-attention (MHSA) to the updated superpixel features. MHSA allows the network to capture long-range dependencies and context information among different superpixels, facilitating a more holistic understanding of the image.

However, we discover that even with multiple iterations $t$ (*e.g.*, $t > 2$) within a SCA module, generating superpixel representations at a lower level may not align perfectly with the overall context,

Table 1: Performance on ImageNet classification. The symbol $^\dagger$ denotes the replacement of the stride-4/8 patch embedding with 2/3 convolution layers, each employing a kernel size of 3.

| Model | #Params | #FLOPs | Top-1 |
|---|---|---|---|
| DeiT-T/16 (Touvron et al., 2021a) | 5M | 1.3G | 72.2 |
| SPFormer-T/16 | 5M | 1.3G | 73.6 |
| SPFormer-T/16$^\dagger$ | 5M | 1.3G | 74.5 |
| DeiT-S/32 (Touvron et al., 2021a) | 22M | 1.1G | 73.3 |
| SPFormer-S/32 | 22M | 1.2G | 76.4 |
| SPFormer-S/32$^\dagger$ | 22M | 1.3G | 77.9 |
| SPFormer-S/56 | 22M | 0.5G | 72.3 |
| DeiT-S/16 (Touvron et al., 2021a) | 22M | 4.6G | 79.9 |
| SPFormer-S/16 | 22M | 5.2G | 81.0 |
| SPFormer-S/16$^\dagger$ | 22M | 5.3G | 81.7 |
| DeiT-B/16 (Touvron et al., 2021a) | 87M | 17.5G | 81.8 |
| SPFormer-B/16 | 87M | 19.2G | 82.4 |
| SPFormer-B/16$^\dagger$ | 87M | 19.2G | 82.7 |

primarily due to the lack of sufficient semantic information. Therefore, we propose a gradual refinement approach for the superpixel representations through multiple SCA modules, where each SCA contains a few iterations (*e.g.*, $t = 2$). The subsequent SCA module will instead exploit the updated pixel features to generate more semantics aligned superpixels.

Specifically, before proceeding to the next SCA module, we project the superpixel features by a $1 \times 1$ convolution and update the pixel features through Eq. 1 with a skip connection (He et al., 2016). The incorporation of information from the globally context-enhanced superpixel features (obtained by previous SCA and MHSA modules) ensures that the pixel representations are refined based on the superpixel context. Instead of initializing the superpixel features from scratch, we use the globally context-enhanced superpixel features from the previous stage as initialization. This process is visualized in Fig. 3. This progressive refinement of the superpixel representations facilitates the capture of increasingly higher-level semantic information.

In the final stage, we apply global average pooling to the superpixel features and feed the resulting representation into a linear classifier for image classification.

Conceptually, our network can be viewed as a two-branch structure: one branch hosts a dense pixel representation with high resolution, and the other accommodates our proposed low-resolution superpixel representation. Minimal operations are directly performed on the dense pixel representation, delegating most of the computation to our efficient superpixel representation. This approach reaps the benefits of computational efficiency, while still preserving local details.

## 4 EXPERIMENTAL RESULTS

In Section 4.1, we provide the implementation details of our proposed method. We then evaluate the efficiency of our approach through a comprehensive ablation study, presented in Section 4.2. Furthermore, we demonstrate the explainability of our method in Section 4.3. Subsequently, in Section 4.4, we showcase its applications in downstream tasks. We further explore its transferability in Section A.1, and robustness in Section A.2.

### 4.1 IMPLEMENTATION DETAILS

Our SPFormer sets a specific relationship between superpixel and pixel feature sizes, where, by design, superpixel features are reduced to $\frac{1}{4} \times \frac{1}{4}$ of their pixel counterparts' spatial dimensions. This downscaling strategy enables the encoding of higher-level, contextually rich information at a more abstract granularity, while maintaining necessary detail in the pixel-level representation. Furthermore, to optimize attention and interaction between superpixels and to facilitate the effective utilization of

global context information, we employ multiple heads for the SCA operation, assigning 2 heads for our tiny and small models, and 3 heads for the base model.

The resulting SCA blocks are inserted into the vanilla ViT architecture just before the first and third self-attention blocks. To stabilize the training process and enhance the convergence of our models, we leverage the LayerScale technique (Touvron et al., 2021b), which is known to aid in maintaining a steady gradient flow during training.

We follow the training recipes (*e.g*., strong data augmentations, AdamW optimizer along with a cosine learning rate scheduler) in DeiT (Touvron et al., 2021a) to train all models on ImageNet (Russakovsky et al., 2015) for 300 epochs. During the training of SPFormer-B/16, we observed a significant issue with over-fitting. In response, we adjusted the Stochastic Depth (Huang et al., 2016) rate from 0.1 to 0.6 to mitigate this problem. We anticipate that more tailored regularization techniques, particularly for the superpixel representation, warrant further exploration in future.

## 4.2 EFFICIENCY

**Main Results**    We first report the evaluation results on ImageNet in Table 1. Our SPFormer can consistently outperform the DeiT baseline across different model sizes and token numbers. For example, under the standard ViT configuration using 196 tokens, our SPFormer-S/16 outperforms DeiT-S/16 by 1.1% (*i.e*., 81.0 *vs*. 79.9%), and our SPFormer-T/16 outperforms DeiT-T/16 by 1.4% (*i.e*., 73.6% *vs*. 72.2%). When transitioning to a larger patchify size of 32, thus reducing the number of tokens, the performance of DeiT-S/32 noticeably declines (from 79.9% to 73.3%). This drop can be attributed to the coarse patch representation's incapacity to adequately capture the entirety of the image. In contrast, our SPFormer-S/32 maintains a highly competitive performance, attaining an accuracy of 76.4%. Notably, this is 4.2% higher than the DeiT-T/16, despite having fewer FLOPs (albeit with more parameters). Moreover, our SPFormer-S/56, which employs only 16 superpixels to represent entire images, achieves an accuracy of 72.3% while operating at a mere 0.5G FLOPs.

An interesting aspect to highlight is that with our SPFormer, the majority of the computational load is borne by the MLPs, differing from typical ViTs where self-attention is usually the main contributor to computations. This observation potentially suggests an alternative scaling rule for our SPFormer—increasing image resolution to capture more fine-grained details. Note that pursuing higher resolutions is not a computationally efficient solution for scaling typical ViTs, as the self-attention complexity in ViT models scales quadratically with the number of tokens. However, our SPFormer can seamlessly adapt to higher resolution inputs, such as 448, by simply increasing the patch size from 4 to 8, all while preserving the same computational cost. By harnessing the details preserved by our superpixel representation without hurting efficiency, SPFormer-S/32 448 achieves an additional improvement of 0.3% over SPFormer-S/16. We refrain from exploring even higher resolutions, as the average resolution of images in the ImageNet dataset is only $469 \times 387$. Conversely, DeiT-S/32 448 derives only a marginal benefit from the increased details, with an improvement of just 0.1% over DeiT-S/16.

Furthermore, in the initial stage of superpixel cross-attention, we utilize a straightforward patchify layer to extract features. This could be further enhanced by introducing a lightweight convolution stem consisting of two or three $3 \times 3$ convolutions with a stride of 2. As reported in Table 1, this modification consistently improves the performance. Remarkably, it further improves the performance of our strong SPFormer-S/32 by an additional 1.5%, attaining 77.9% ImageNet accuracy.

**Ablation Study**    We investigate the design choices of our proposed method by evaluating SPFormer-S/32 on the ImageNet validation set, as summarized in Table 2. Traditional superpixel methods often require multiple iterations to converge (Achanta et al., 2012; Jampani et al., 2018). Similarly, in our approach, using only one iteration within the Superpixel Cross Attention (SCA) block leads to a degradation in performance by 1.0%. This emphasizes the importance of iteratively refining the superpixel features. Furthermore, inserting the SCA at only the initial (0th) layer in the Vision Transformer (ViT) architecture results in a decrease in top-1 accuracy by 1.6%. This observation underscores the significance of gradually incorporating and refining the superpixel features throughout the network, as lower-level features lack sufficient semantic information.

Given the inherent ambiguity in decomposing images into multiple superpixels, the absence of our multi-head SCA design leads to a performance drop of 0.8%. This demonstrates the necessity of

leveraging multiple attention heads to capture diverse relationships between superpixels. Lastly, the substitution of the CPE with learnable position embeddings results in a performance drop of 0.3%.

Overall, these ablation studies verify the effectiveness of our proposed design choices, such as iterative refinement, strategic placement of the SCA module and multi-head cross-attention, in achieving superior performance in image classification tasks.

Table 2: Ablation study on the design choices.

| Model | #Params | #FLOPs | Top-1 |
|---|---|---|---|
| SPFormer-S/32 | 22M | 1.2G | 76.4 |
| iteration $t = 1$ | 22M | 1.2G | 75.4 |
| first stage only | 22M | 1.2G | 74.8 |
| single head SCA | 22M | 1.2G | 75.6 |
| learnable position embedding | 22M | 1.2G | 76.1 |

### 4.3 EXPLAINABILITY

The superpixel representation itself can be understood through the visualization of the association matrix. In Figure 1, we directly visualize the learned soft association $S_a$ by selecting the argmax over the superpixels using the following equation:

$$\hat{S_a} = \mathrm{argmax}(S_a) \tag{5}$$

By analyzing these visualizations, we observe that even when trained with a soft association, the superpixels generally align with the boundaries in the image. This behavior is particularly interesting since the network is only trained using image category labels. Despite drastically reducing the number of tokens required to represent the image, the learned superpixels can effectively segment the images into irregular regions with semantics-aware.

To rigorously evaluate the quality of superpixels, we perform a quantitative assessment of their edge alignment with the groundtruth at both object and part levels, utilizing the Pascal VOC 2012 dataset (Everingham et al., 2015) and Pascal-Part-58 (Zhao et al., 2019). For each superpixel or patch, we make predictions by aggregating the labels assigned to the pixels within it based on the ground truth. Specifically, we assign the label that occurs most frequently within a superpixel to its corresponding prediction, assuming perfect superpixel or patch classification. Given that our Superpixel Cross Attention (SCA) module produces soft associations, our superpixel predictions are generated by combining pixel labels with appropriate weights. These weighted predictions are then upsampled using Eq. 1.

In contrast to patch representation and traditional superpixel methods, which yield a single superpixel, our approach employs a multi-head design, generating distinct superpixels for each head, as depicted in Figure 4. For simplicity in evaluation, we calculate an average of the predictions from all heads. It is important to note that during the feature extraction process, successful extraction occurs if at least one head correctly identifies a superpixel.

As illustrated in Table 3, vanilla ViTs adopt a patch representation with a stride of 16, which potentially leads to compromised performance due to its coarser granularity. In contrast, the superpixels obtained through our SCA module demonstrate substantial improvements, yielding an increase of 4.2% in object level and 4.6% in part level mean Intersection over Union (mIoU) in our SPFormer-S[†]. Furthermore, our superpixels exhibit a quality level comparable to those generated by the traditional superpixel method, such as SLIC (Achanta et al., 2012).

### 4.4 SEMANTIC SEGMENTATION

Given the inherent capability of superpixel representation to preserve finer details compared to patch representation, our approach is particularly well-suited for tasks demanding dense predictions, such as semantic segmentation. We follow the methodology outlined in SETR (Zheng et al., 2021), which involves a straightforward bilinear upsampling of the patch representation. Building on the strengths

Table 3: Zero-shot evaluation of superpixel quality. 196 patches/superpixels are used.

| Method | Pascal Voc2012 | | Pascal-Parts-58 | |
|---|---|---|---|---|
| | mIoU | mAcc | mIoU | mAcc |
| Patch | 87.8 | 92.8 | 68.7 | 78.2 |
| SPFormer-T/16[†] | 91.5 | 95.7 | 71.5 | 79.9 |
| SPFormer-S/16[†] | 92.0 | 96.6 | 73.3 | 82.4 |
| SPFormer-B/16[†] | 91.2 | 96.3 | 72.5 | 81.4 |
| SLIC (Achanta et al., 2012) | 92.5 | 95.4 | 74.0 | 81.7 |

of our superpixel representation, we proceed with the direct classification of individual superpixels. Subsequently, we upsample the associated logits using Eq. 1.

We conduct evaluations on both the ADE20K dataset (Zhou et al., 2017) and the Pascal Context dataset (Mottaghi et al., 2014). For these evaluations, we utilize input resolutions of $512 \times 512$ and $480 \times 480$ for ADE20K and Pascal Context, respectively. In our training recipe, we employ AdamW with an initial learning rate of $6 \times 10^{-5}$ and a weight decay of $10^{-2}$. Training is conducted for 160k/40k iterations for ADE20K/Pascal Context, respectively, with a batch size of 16.

Utilizing pretrained models on ImageNet, our method exhibits a performance gain of 4.2% and 2.8% in mean Intersection over Union (mIoU) on the ADE20K and Pascal Context datasets, respectively. In order to isolate the impact of a superior pretrained model and to further understand the benefits of incorporating the superpixel representation, we conduct additional training from scratch in both scenarios. Here, our method surpasses the baseline by 3.0% and 3.1% in mIoU for ADE20K and Pascal Context, respectively, reaffirming the advantages conferred by our superpixel representation.

Table 4: Semantic segmentation on ADE20K val split.

| Method | #Params | #FLOPs | Imagenet Acc | mIoU |
|---|---|---|---|---|
| DeiT-S/16 | 22M | 32G | - | 20.1 |
| SPFormer-S/16 | 23M | 35G | - | 23.1 |
| DeiT-S/16 | 22M | 32G | 79.9 | 42.3 |
| SPFormer-S/16 | 23M | 35G | 81.0 | 46.5 |

Table 5: Semantic segmentation on Pascal Conext val split.

| Method | #Params | #FLOPs | Imagenet Acc | mIoU |
|---|---|---|---|---|
| DeiT-S/16 | 22M | 27G | - | 18.0 |
| SPFormer-S/16 | 23M | 30G | - | 21.1 |
| DeiT-S/16 | 22M | 27G | 79.9 | 48.3 |
| SPFormer-S/16 | 23M | 30G | 81.0 | 51.2 |

## 5 CONCLUSION

The vision community has observed the change of feature representation learning from pixel representation (for convolution neural networks) to patch representation (for vision transformers). In this work, we propose a novel superpixel representation to enhance vision transformers. As a result, the proposed SPFormer (Superpixel Transformer) provides three benefits: efficiency (the small number of superpixels is amenable to global-range self-attention), explainability (the learned superpixel groups pixels with similar semantics), and robustness (the learned superpixel is adjustbable to image distortion) that existing pixel or patch representation could not all satisfy. Our results on the challenging ImageNet related datasets have shown encouraging results. We hope our work will inspire future research on this promising feature representation.

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

# A APPENDIX

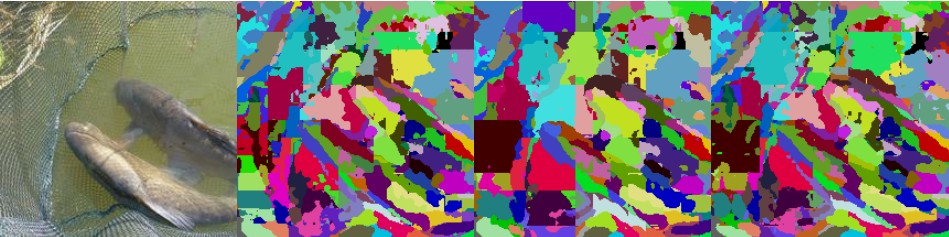

Figure 4: The multi-head design of SCA results in the generation of multiple superpixels.

## A.1 TRANSFERABILITY

Figure 5 showcases the visual representation of superpixels generated on the COCO dataset (Lin et al., 2014). For evaluation purposes, we resize the high-resolution COCO images and apply a center crop, adhering to the ImageNet evaluation pipeline. Remarkably, we observe that the superpixel representations exhibit a degree of generalization to unseen data. Interestingly, it captures the thin structure like the fork and straw, thanks to the detail preserved through our superpixel representation. It is important to note that our models are exclusively trained on the ImageNet dataset using category labels. Quantitative evaluations are detailed in Table 3.

## A.2 ROBUSTNESS

**Robustness to Rotation** Figure 6 illustrates the ability of our model to generate reasonable superpixels even under rotation. We observe that, leveraging the superpixel representation, SPFormer demonstrates increased robustness to rotation, as evidenced in Table 6. However, our model still exhibits some susceptibility to rotation, despite employing a somewhat rotation-invariant superpixel representation. We suspect that this limitation arises from the fact that the learnable absolute position is not inherently rotation-invariant.

These findings highlight the potential for further exploration to improve the rotation robustness of our model by considering rotation-invariant mechanisms in the superpixel representation or incorporating explicit rotational invariance in the network architecture.

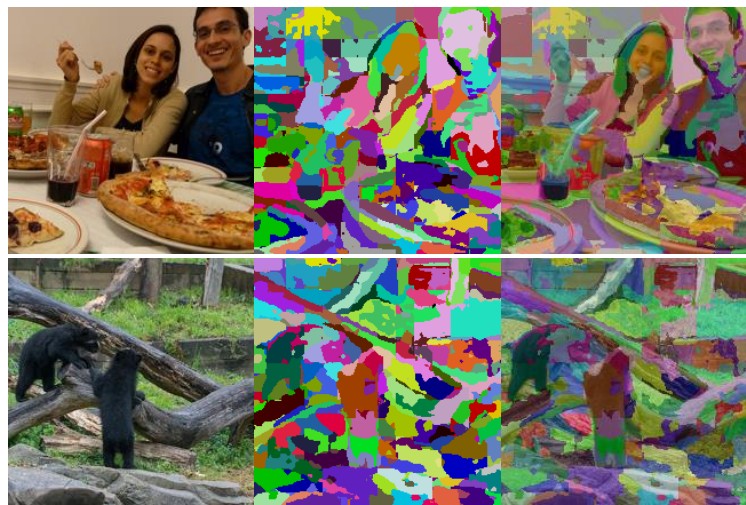

Figure 5: Zero-shot transferability on COCO dataset. The model is exclusively trained on the ImageNet dataset using category labels. The superpixels are overlaid onto the images to enhance visualization and facilitate understanding. 196 superpixels are used.

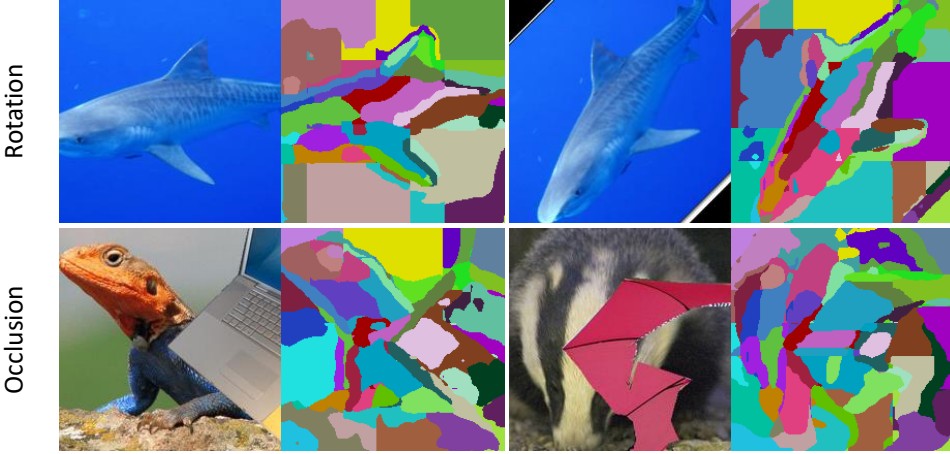

Figure 6: Illustration of the robustness to the rotation and occlusion of the superpixel representations.

**Robustness to Occlusion** In Figure 6, we present visual results for images with occlusion, demonstrating the effectiveness of our superpixel representations. Our superpixel representations exhibit the ability to differentiate between the object and occluders to some extent. In contrast, patch representations inherently mix the object and occluders within some patches, potentially making it more challenging for the network to generate accurate features.

Table 6: Robustness to rotation.

| Model | Clean | 15 | 30 | 45 |
|---|---|---|---|---|
| DeiT-S/32 (Touvron et al., 2021a) | 73.3 | 71.1 | 67.7 | 59.5 |
| SPFormer-S/32[†] | 77.9 | 75.2 | 73.4 | 66.9 |

## A.3 COMPARISION WITH SViT

We conducted a comparative analysis between our superpixel representation and that of SViT (Huang et al., 2023) in both non-hierarchical and hierarchical settings, as presented in Table 7. In the non-hierarchical setting, substituting our superpixel representation with SViT's led to a notable drop in performance. Furthermore, enabling gradient flow resulted in training instability, with occurrences of NaN values. We hypothesize that this might be the reason why SViT opts to disable gradient flow, despite the differentiability of its soft association.

In the hierarchy setting, we start from the architecture of SViT, which contains four stages. SViT incorporates super token sampling in each layer of the first two stages, performing self-attention operations between super tokens before subsequently upsampling them back to the pixel space. Following this, ConvFFN is applied to process the pixel features. As discussed in Sec. 3.3, we redirect all computations to the superpixel representation as depicted in Fig. 3, denoting this variant as SViT$^*$. Note that the last two stages are the same in SViT and SViT$^*$. As demonstrated in Tab.7, SViT$^*$ exhibits a slight performance improvement over SViT[2], while requiring fewer FLOPs, which highlights that computations on pixel features are expensive. The utilization of our formulation leads to a 0.3% enhancement in performance. It's important to highlight that SViT employs a small number of super tokens (49), resulting in a relatively low computational cost of around 50 million FLOPs for the ViT blocks. To further optimize performance, we increase the dimension of superpixels akin to the approach proposed by Sandler et al.(Sandler et al., 2018), resulting in an additional 0.3% improvement.

Table 7: Compare with SViT.

| Method | FLOPs | Params | Acc |
|---|---|---|---|
| Non-Hierarchy | | | |
| SPFormer-S/32 | 1.2G | 22M | 76.4 |
| +SViT style | 1.1G | 22M | 68.5 |
| +SViT style with grad | 1.1G | 22M | NaN |
| Hierarchy | | | |
| SViT | 4.4G | 25M | 80.7 |
| SViT$^*$ SViT | 3.3G | 25M | 80.8 |
| + expansion 2 | 3.5G | 29M | 80.8 |
| SViT$^*$ Our | 3.5G | 26M | 81.1 |
| + expansion 2 | 3.7G | 29M | 81.4 |

---

[2]The SViT result is obtained by running the officially released code

