# OpenReview forum: "SPFormer: Enhancing Vision Transformer with Superpixel Representation"
_ICLR.cc/2024/Conference — ICLR 2024 Conference Withdrawn Submission_

### Official Review · Reviewer_C59i · 2023-10-30

**Soundness:** 2 fair
**Presentation:** 2 fair
**Contribution:** 2 fair
**Rating:** 3
**Confidence:** 4

**Summary:**

The paper proposes SPFormer, which is a vision transformer using a superpixel representation instead of a patch-based representation.
The core module consists of two attention blocks, pixel-to-superpixel cross-attention and superpixel-to-pixel cross-attention, which alternately update the superpixel and pixel features.
In the experiments, SPFormer shows better performance on ImageNet-1k, compared to DeiT which is a patch-based baseline.

**Strengths:**

- The paper shows the superpixel representation is superior to the patch representation in a ViT architecture.

**Weaknesses:**

1. Incorporating superpixels into the neural nets has been well studied, and the essential difference between the proposed module and previous studies is updating the pixel feature. However, the proposed module is not compared with other clustering modules (e.g., SSN[1] and GroupViT), and the advantage of the proposed module is not verified.

2. The authors discussed the existing architecture using superpixel representation, but the authors compare the proposed method only with patch-based architecture, and there is no comparison in the experiments. In fact, the accuracy of the proposed method on ImageNet-1k is worse than that of SViT[3].

3. I wonder how useful explainability is. Visualizing superpixels just shows similar pixels and does not explain why the model makes the decision. The similarity can also be visualized by clustering pixels in post-processing.

[1] Jampani, Varun, et al. "Superpixel sampling networks." 2018
[2] Xu, Jiarui, et al. "Groupvit: Semantic segmentation emerges from text supervision." 2022
[3] Huang, Huaibo, et al. "Vision Transformer with Super Token Sampling" 2023

**Questions:**

- The author mentioned that MHSA allows the network to capture long-range dependencies and context information among different superpixels in Sec. 3.3. However, SCA adopts the sliding window-based cross-attention, and I think it captures only local dependencies. Could you clarify it?

---

### Official Review · Reviewer_mr9V · 2023-10-31

**Soundness:** 3 good
**Presentation:** 3 good
**Contribution:** 2 fair
**Rating:** 3
**Confidence:** 5

**Summary:**

This paper introduced a new attention mechanism that cross attend superpixel and pixel features for vision transformers. The proposed SPFormer approach tried to achieve efficiency and robustness by first segment the image into superpixels and build ViTs in a cross attention formulation. As a result, the formulation achieved decent result for image classification and segmentations tasks on public benchmarks.

**Strengths:**

- I think the visualizations of this paper is decent enough for people to follow
- The use of superpixels is an interesting idea, and I think it's good to see people trying out this approach

**Weaknesses:**

- Experiments are done on imageNet datasets for image classification tasks, and only compared with relatively old approaches like DeiT. Most state-of-the-art approach is using Swin-like shifting window attention mechanisms, and the reported numbers are relatively low in year 2023. It's really challenging to justify the effectiveness of this approach.
- The main motivation of using super pixel representations is (as stated by the authors): 1) efficiency, and 2) can potentially handle high resolution images. However the paper doesn't report any speed numbers or tried evaluating on images with high resolutions (and comparisons with other approaches). It's again very challenging to justify the effectiveness of this approach.

**Questions:**

Looks like the supplementary material is identical to the main paper, maybe there is an issue?

---

### Official Review · Reviewer_BcpC · 2023-11-01

**Soundness:** 3 good
**Presentation:** 2 fair
**Contribution:** 2 fair
**Rating:** 5
**Confidence:** 4

**Summary:**

The paper suggests SPFormers, an extension to ViTs that proposes to use superpixels instead of pixels. This modification has the appeal of reducing the spatial extent of the inputs which drastically reduces the number of tokens, this favorably affects memory requirements as attention is typically quadratic in the number of tokens.

**Strengths:**

The idea to combine superpixels with transformer architectures sounds interesting, especially since transformers are prone to high memory requirements
The introduction manages to explain the problem and the appeal of the solution
The related work section is extensive and well-written

**Weaknesses:**

Section 3.1.: There are a couple of things that are confusing. First of all, the notation is unorthodox, with ph/pw (or sh/sw) the authors probably mean subscripted p_h, p_w rather than its multiplicative interpretation. Equation (1) is also confusing as it is unclear what “p” really denotes. Assuming that “p” is a spatial coordinate (e.g. (x, y)), we run into the problem that the last dimensions of S_f are sh x sw, while they are h x w for S_a. This should be made clearer, e.g. with the help of an illustration as this deviates from the “common” definition of superpixels. It should also be defined how the “n” dimension is to be interpreted, is it a one-hot dimension?
Unfortunately, the latter weakness makes it very difficult to follow the rest of the paper as the next equations become very specific without providing much intuition. The illustrations are only partially helpful.
SPFormer is only compared against DeIT (and S-ViT in the Appendix). Comparing the model against a single model in the main paper is insufficient and would suggest adding more recent models (more recent DeIT derivates or Swin transformers).
Given that SPFormers are claimed to be a more efficient model class it would have been interesting to see how well they perform if the resolution is increased. This could be done in comparison to other transformer models that would be expected to perform worse
Typo(s): “adjustbable” in the conclusion

**Questions:**

Have you considered comparing against zero-shot foundation segmentation models such as SAM or SEEM?

---

### Official Review · Reviewer_JGdc · 2023-11-07

**Soundness:** 3 good
**Presentation:** 2 fair
**Contribution:** 3 good
**Rating:** 5
**Confidence:** 4

**Summary:**

This paper proposes an enhancement to the vanilla vision transformer (ViT) by substituting the patch representation for superpixel representation. The superpixel representation is constructed iteratively by multiple Pixel-to-Superpixel and Superpixel-to-Pixel cross attention modules. With this superpixel cross attention (SCA) module, the proposed SPFormer architecture is composed of stages which by turns optimize local associations between pixels and superpixels and capture long-range dependencies between superpixels. The enhancement made to the vanilla vision transformer provides higher efficiency, more explainability and more robustness.

**Strengths:**

* The proposed SPFormer introduces an innovative way of jointly optimizing superpixel and pixel representations. The converged superpixels are naturally aligned semantically
* The proposed SPFormer introduces efficient superpoint representation which significantly lower the resource requirements
* The proposed SPFormer outperforms the chosen baselines by significant margins

**Weaknesses:**

* The relationships between pixels and superpixels are poorly explained. Specifically, the neighboring relationship described by "pixels surrounding superpixel" and "neighboring superpixels" are ambiguous. As the interaction between pixels and superpixels is a foundamental part of this work, this ambiguity severely lower the quality of the otherwise significant work.
* The comparison with SViT (Huang et al., 2023) seems problematic. According to the original paper of Huang et al., the accuracy of SViT on ImageNet is higher than 83.6% whereas the accuray of SViT reported in this work is 80.7%. This significant difference should be better explained.
* The experiments for transferability are limited. According to Table. 3, The zero-shot ability is only demonstrated at the level of the superpixel/patch quality, compared with relatively weak baselines patch and SLIC (Achanta et al., 2012). Further experiment results including the zero-shot ability of SPFormer itself compared with SViT and other competitive baselines.

**Questions:**

See the part above (Weaknesses).